# Attitudes and barriers to exercise in adults with a recent diagnosis of type 1 diabetes: a qualitative study of participants in the Exercise for Type 1 Diabetes (EXTOD) study

Amy Kennedy,[1] Parth Narendran,[2] Robert C Andrews,[3] Amanda Daley,[4] Sheila M Greenfield,[4] for the EXTOD Group

[1]The Institute of Metabolism and Systems Research and Centre for Endocrinology, Diabetes and Metabolism, The Medical School, University of Birmingham, Birmingham, UK
[2]The University of Birmingham and The Queen Elizabeth Hospital, Birmingham, UK
[3]Institute of Health Services Research, University of Exeter Medical School, University of Exeter, Exeter, UK
[4]Institute of Applied Health Research, University of Birmingham, Birmingham, UK

**Correspondence to**
Dr Robert C Andrews;
r.c.andrews@exeter.ac.uk and
Dr Parth Narendran;
p.narendran@bham.ac.uk

## ABSTRACT

**Objectives** To explore attitudes and barriers to exercise in adults with new-onset type 1 diabetes mellitus (T1DM).

**Design** Qualitative methodology using focus group (n=1), individual face-to-face (n=4) and telephone interviews (n=8). Thematic analysis using the Framework Method.

**Setting** Nineteen UK hospital sites.

**Participants** Fifteen participants in the Exercise for Type 1 Diabetes study. We explored current and past levels of exercise, understanding of exercise and exercise guidelines, barriers to increasing exercise levels and preferences for monitoring of activity in a trial.

**Results** Five main themes were identified: existing attitudes to exercise, feelings about diagnosis, perceptions about exercise consequences, barriers to increasing exercise and confidence in managing blood glucose. An important finding was that around half the participants reported a reduction in activity levels around diagnosis. Although exercise was felt to positively impact on health, some participants were not sure about the benefits or concerned about potential harms such as hypoglycaemia. Some participants reported being advised by healthcare practitioners (HCPs) not to exercise.

**Conclusions** Exercise should be encouraged (not discouraged) from diagnosis, as patients may be more amenable to lifestyle change. Standard advice on exercise and T1DM needs to be made available to HCPs and patients with T1DM to improve patients' confidence in managing their diabetes around exercise.

**Trial registration number** ISRCTN91388505; Results

## BACKGROUND

Regular physical activity plays a key role in the management of patients with type 1 diabetes mellitus (T1DM). It improves insulin sensitivity, reduces cardiovascular risk factors such as blood pressure (BP) and lipid profiles, improves quality of life and reduces mortality.[1] As a result, patient guidelines currently recommend undertaking at least 150 min per week of moderate to vigorous aerobic exercise, spread out during at least 3 days, with

## Strengths and limitations of this study

► This is the first qualitative interview study to examine attitudes and barriers to exercise in patients newly diagnosed with type 1 diabetes mellitus.
► Patient recruitment was from UK sites covering both large teaching and district general hospitals and participants spanned a wide age range.
► Study participants may have been more interested in exercise than those who declined and interest in exercise education and management of diabetes around exercise may be lower in the general clinic population.

no more than two consecutive days between bouts of aerobic activity. Patients should also be encouraged to perform resistance exercise 'at least two times per week on non-consecutive days'.[2 3]

A large percentage of patients with T1DM do not reach these guidelines. In a retrospective analysis of the Diabetes and Complications Trial, 19% of (271/1441) participants were not achieving American Diabetes Association (ADA) activity level recommendations.[4] In the EURODIAB prospective cohort study of 2185 patients with T1DM from 16 European countries, 786 (36%) patients were doing none or only mild physical activity.[5] Similarly 23% of patients with T1DM were classed as sedentary and a further 21% were doing less than one session of exercise per week in the Finnish Diabetic Neuropathy Study.[6]

Little is known about attitudes and barriers to exercise in patients with T1DM. In two Canadian studies of patients with established T1DM,[7 8] fear of hypoglycaemia was the strongest barrier to regular exercise. A qualitative study from our group in the UK suggests that although fear of hypoglycaemia is a factor

when patients with established T1DM consider exercise, external factors, such as lack of time, work pressures and bad weather were greater barriers to physical activity.[9]

No studies have examined attitude and barriers to exercise in patients recently diagnosed with T1DM, a time when exercise habits may be greatly influenced. This qualitative study aimed to explore attitudes and barriers to exercise in adults with new-onset T1DM.

## METHODS
### Recruitment
Study patients were from the EXercise for Type 1 Diabetes study (EXTOD) whose protocol has been described previously.[10] In brief, all patients aged between 16 and 60 years, diagnosed with T1DM in the previous 3 months from 19 UK hospital sites were invited to participate. EXTOD had two phases, Phase 1 which consisted of the qualitative study reported here. This was designed to inform on the most feasible and patient-friendly way of motivating patients newly diagnosed with T1DM to undertake and maintain a graded exercise programme and to determine attitudes and barriers to exercise. This understanding was essential for the conduct of Phase 2, a pilot randomised controlled trial to assess uptake, intervention adherence, dropout rates and rate of uptake in the usual care group during a 12-month exercise intervention (not the subject of this report). Participants were approached by a member of the clinical team (doctor/diabetes nurse/dietitian) at their local site and gave written informed consent.

### Interviews
Initially it was intended to use focus groups but geographical spread and the time interval between identification of participants meant one to one and telephone interviews had also to be offered.

Interviews were carried out by AK, using a semistructured topic guide,[10] and lasted between 30 and 60 min. Areas for discussion included current and past levels of exercise, understanding of exercise and exercise guidelines, barriers to increasing exercise levels and preferences for monitoring of activity in a trial.

### Analysis
Interviews and focus groups were recorded and transcribed. Data analysis was ongoing during the collection period to enable full exploration of themes identified in earlier interviews and to identify when saturation had been achieved.[11] Data were managed using N-Vivo 9 (QSR International, Victoria, Australia). Themes and a coding frame were developed independently by reading and re-reading interview transcripts and through discussions between research team members (AK, PN and SG). Interviews were then analysed using a framework approach to further examine identified themes.[12]

## RESULTS
### Participants
Fifteen participants were interviewed: one focus group of three participants, four face-to-face and eight by telephone (table 1). Eleven were male, median age was 29 (range 18–53 years) and 12 were of White-British ethnic origin. The median length of time from diagnosis to interview was 66 days.

### Themes
The interviews yielded rich data on five main themes. These were: exercise context (attitudes to and current

**Table 1** Participant demographics

| Participant | Age group | Gender | Centre | Ethnic origin | Interview format | Group |
|---|---|---|---|---|---|---|
| A | 40–44 | m | Bir | Asian or Asian British—Indian | FG, Face-to-face | CONCERNED |
| B | 20–24 | f | Bir | White—British | FG, Face-to-face | CONCERNED |
| C | 50–54 | m | Bir | White—British | FG, Face-to-face | CONCERNED |
| D | 50–54 | m | Bir | Black or Black British—Caribbean | I, Face-to-face | CONCERNED |
| E | 20–24 | m | Bir | White—British | I, Face-to-face | CONFIDENT |
| F | 35–39 | m | Tau | White—British | I, Face-to-face | AMBIVALENT |
| G | 20–24 | m | Glou | White—British | I, Face-to-face | AMBIVALENT |
| H | 20–24 | m | Brist | White—British | I, Telephone | CONFIDENT |
| I | 50–54 | m | Bir | White—British | I, Telephone | CONCERNED |
| J | 20–24 | f | Wake | White—British | I, Telephone | CONFIDENT |
| K | 45–49 | f | Glou | White—British | I, Telephone | CONFIDENT |
| L | 15–19 | m | Bir | White—British | I, Telephone | AMBIVALENT |
| M | 35–39 | m | Tau | Mixed—White and Black African | I, Telephone | CONCERNED |
| N | 25–29 | f | Bir | White—British | I, Telephone | CONCERNED |
| O | 15–19 | m | Brist | White—British | I, Telephone | CONFIDENT |

FG, focus group; I, individual interview.

| Participant | Activities prior to diagnosis | Current activities |
|---|---|---|
| A | Jogging, rope skipping, playing football | Walking while at work (4–5 hours a day) |
| B | Walking at work, gardening, do it yourself (DIY) jobs, gym, squash | Occasional gym session, DIY |
| C | Physical job, gardening, DIY, repairs | None |
| D | Regular attendance at the gym (cardiovascular and weight training) | Walking |
| E | Marshall arts/boxing | Active job 2 days a week |
| F | Walking while at work | Walking while at work |
| G | Swimming, jogging | Swimming, jogging |
| H | Combat karate | Jogging, some weights |
| I | Walking/jogging outside | Walking on treadmill |
| J | Gym | Gym |
| K | Gardening | Gardening, walking |
| L | Walking | Walking |
| M | Running | Running |
| N | Rugby, football, cycling | Cycling on static bike |
| O | Badminton/golf | Badminton and golf |

Table 2  Activities described as exercise by participants

and previous exercise behaviour); diabetes (impact of diagnosis and knowledge); consequences of exercise; barriers to increasing exercise; confidence (in exercising and managing diabetes).

Specific numbers of participants are not routinely given throughout as these are not generally used when reporting qualitative research, the aim of sampling being to represent the spread of views rather than proportions which can be generalised to a larger group.[13]

### Attitudes to and current and previous exercise behaviour

All participants were already doing some form of exercise with the majority wanting to increase activity levels. Activities that participants classed as exercise varied from walking during their working day to swimming or going to the gym. Table 2 shows the exercise that each participant was taking part in. Ten participants were doing moderate activity, one moderately vigorous activity and three vigorous activity. Five participants reported a reduction in the amount of time they spent exercising, and seven had changed the type or reduced the intensity of activities they were doing since diagnosis. Most participants were either unaware there is guidance on the minimum amount of exercise adults should undertake each week or uncertain as to the amount recommended. Many were pleasantly surprised recommendations were not higher and felt they should be able to achieve this even if they were not already doing so. Some felt a universal guideline was inappropriate as it could not include individual circumstances and a personalised target would be preferable.

'Because each person should be done individually. And the doctor should say yes, you're capable of doing this. No, you're not …because he'll have your medical records, …Not the government

telling you, you should do this or you should do that.' (Participant C).

### Impact of diagnosis and knowledge of diabetes

All participants talked about the impact of their T1DM diagnosis, most commonly describing the sudden nature of the diagnosis of as a '*shock*' (A, D, H, I, K, M and N). Other descriptions were as being '*hit*', a '*kick in the teeth*' (both participant C) and feeling '*stunned*' (participant I). Several participants described their diagnosis as a loss of normality (wanting to get back to a 'normal life') or role (uncertainty about being able to work).

Participants reported four different fears and anxieties regarding their T1DM diagnosis: managing new interactions with healthcare services; impact on employment, concerns for the future and blood glucose control. Some reported feeling overwhelmed by the amount of contact they had with healthcare services since diagnosis.

'Every other week I'm getting different, another letter through with different things which could be related to it' (Participant D)

'there's too many things going on at the moment, I think for me.' (Participant K)

For several participants, T1DM had negatively impacted on work. Some had still not gone back to work and were anxious about their ability to cope. One (N) had lost their job.

'I'm quite concerned about going back to work actually. Because I know that I'm going to be on the go all the time and whether I'm going to be able to cope with doing 8 hours worth of walking on a daily basis'. (Participant B)

'That's the problem, going back into a job now, knowing if you can do it.' (Participant C)

Some participants had concerns for the future and reported uncertainty about their future health. One participant had discussed this with their general practitioner (GP).

'I goes to him [the GP] 'how long are you going to live on it?' He goes 'if you don't look after yourself, he says, 5 years'. I thought, what! That's a serious thing.' (Participant D)

'it's just nobody has sort of come out and said like, 'This is exactly like, you know, what's going, what's going to happen and stuff like that.' (Participant F)

Some participants were concerned about blood glucose levels and many were anxious to get optimal glycaemic control. Participants expected their blood glucose levels would become 'balanced' with time and they would then be able to keep them within a tight range.

Importantly, all said being diagnosed with diabetes had given them additional motivation to exercise than before diagnosis (even those who did not plan to increase activity levels).

'it's changed my ethos of taking time to do some exercise in some, you know, going for walks. It's changed my mind, my what I think.' (Participant K)

'I mean generally the reason most diabetics start, or people in general start doing more exercise is because of the fear. At the end of the day I think it's the fear factor of being afraid that if I don't then my life is going to be worse.' (Participant E)

Twelve participants wished to increase activity levels, although some had more concrete plans than others.

'but actually, I could do my 10 min [bout of exercise], because we do have a room that nobody ever goes into, erm, so I could do that here, and that's a thought, maybe I could consider.' (Participant M)

### The consequences of exercise
Perceptions about the consequences of exercising were mostly positive and included; health benefits, improved fitness, enjoyment, a feeling of well-being and weight loss. Some participants cited exercise benefits specifically related to diabetes such as lower blood glucose and insulin requirements.

Although health benefits were commonly mentioned as a motivation to exercise, often participants were vague about them and unable to give specific examples. A few mentioned positive effects on BP, cholesterol and heart disease risk.

Blood glucose lowering was seen to be a positive effect of exercise by some, for others this was a negative result as it was associated with hypoglycaemia. Those participants were particularly concerned about hypoglycaemia and whether this would counteract the health benefits of

exercise, both directly as a consequence of hypoglycaemia and also secondary to the need to increase carbohydrate intake.

Participant C in particular felt there was little point in exercising as although he had previously been active, this had not prevented him developing T1DM.

'all of a sudden they get diabetes, and they say you've got to have insulin, then they say you've got to exercise to reduce your insulin. Well hang on, I've been exercising all my life, and why have I got to end up taking insulin?' (Participant C)

### Barriers to exercise
Two main subthemes emerged, medical barriers and the influence of healthcare practitioners (HCPs). In addition, individual barriers to increasing exercise mentioned by participants were noted (table 3).

### Medical barriers to exercise
Most medical factors were diabetes-related. Most frequently cited was hypoglycaemia (nine participants). For some, this related to actual experience of hypoglycaemia during or after exercise, others were worried about hypoglycaemia but had not yet experienced this. Seven participants cited lack of knowledge or confidence in managing diabetes around exercise. Four people mentioned the need to plan for exercise with diabetes, for example, checking blood glucose before and during activity and preparing for hypoglycaemia, as a discouraging factor. Fatigue (which may be related to hyperglycaemia) was cited by four people. Three people talked about other aspects of physical health being a barrier to exercise; all had experienced an injury.

### Influence of healthcare practitioners
HCP advice could be either positive or negative. Four participants said HCPs had advised them not to exercise.

'They advised me to do no exercise basically at the hospital until they felt like I could.' (Participant B)

'Because I was asking in the hospital, I kept going, have you got a gym here? 'oh, you've got diabetes, you can't be going to the gym' and stuff like that.' (Participant D)

Some participants (who were successfully exercising) described how helpful and supportive (of exercise) they had found HCPs.

'I was a bit cautious, erm, about, erm, doing anything to start [laughs] with, really, but I spoke to the nurses and they were just, you know, within reason, they just said, 'Carry on your life as normal,' really' (Participant N)

'because when I asked about the fact that I go running, 'Yeah, that's brilliant. That's great,'' (Participant M)

**Table 3** Barriers to increasing exercise cited by participants

| External | Barrier (number of people mentioning barrier) |
|---|---|
| Medical | Hypoglycaemia (both actual and fear of) (9) |
| | Lack of knowledge/confidence in managing diabetes (6) |
| | Fatigue (4) |
| | Advice from healthcare professionals to stop exercising (4) |
| | Planning for diabetes (eg, checking blood glucose/preparing for hypoglycaemia) (4) |
| | Other physical health problems (eg, injuries) (3) |
| | Feeling overwhelmed by diagnosis. (1) |
| Time, work and environmental | Work commitments (9) |
| | Family and other time commitments (6) |
| | Availability and location of facilities (4) |
| | Cost (4) |
| | Weather/season (3) |
| | Lifestyle (2) |
| Internal Social and personal | Lack of fitness (3) |
| | Lack of motivation (2) |
| | Lack of enjoyment in certain activities (2) |
| | Laziness (1) |
| | Previous negative experience of exercise (1) |
| Psychological | Feeling uncomfortable exercising (eg, at a gym) (2) |
| | Feeling scared of exercising on own (2) |
| | Feeling daunted at prospect of starting (2) |

However, one participant although generally positive about HCP support, did comment that this was not routinely offered.

'my team have been brilliant with me so far, and [exercise is] perhaps something I haven't remembered necessarily to ask when I'm there, but at the same time I'm not sure it's offered that freely.' (Participant N)

Several participants thought they had been given conflicting advice about exercise and diabetes and felt some HCPs were not well informed about T1DM. Participants found this frustrating.

'because it seems like, you know, everybody seems to have slightly different things to say about it, whoever I ask.' (Participant H)

'I also have a problem though, that you've got doctors in a hospital telling one thing to you, not the diabetic team, another doctor telling you you're type 2.' (Participant K)

Importantly, participants who reported doing most activity (J, K and O) were among the group who had had positive experiences. Conversely, participants who reported doing no exercise at all (C, D, H) said they had either been told not to exercise or received conflicting advice.

### Individual barriers to exercise

Twenty-one different barriers to increasing exercise levels were mentioned (table 3), most commonly hypoglycaemia and work commitments (nine participants). Barriers fell into four categories, either external (medical, time, work and environment) or internal factors (social and personal, psychological). Participants tended to cite a variety of external factors, with only a few discussing internal barriers.

### Confidence in exercising and managing diabetes

Participants' confidence both in their ability to perform activities and manage their blood glucose around exercise was a major factor influencing determination to increase exercise levels.

When considering confidence, participants described three areas: managing diabetes, exercising and managing diabetes around exercise.

Some participants felt they had little control over their diabetes or that something had knocked their confidence, whereas others had developed or maintained confidence in their ability to cope with blood glucose fluctuations.

'because I've had this problem where everything has gone a bit odd, for the last couple of weeks, I think it's set me back a bit and perhaps I want to be more confident, I want to make sure I've got my background insulin right' (Participant K)

I'm a lot more aware of being out on my- even just being out on my own, especially at the beginning, sort of, if I was asked to babysit and I, kind of, went, 'Oh, are you sure you trust me? What if something happens to me?' (Participant N)

Some participants lacked confidence in exercising prior to diagnosis, others were not sure if there were any special considerations due to their diagnosis.

'I was never good [at exercise] at school' (Participant M)

Other participants discussed their confidence in exercising now they had been diagnosed with diabetes.

'my confidence is, I at the moment, I've had a couple of sessions when I've been doing gardening and I've said oh, my legs feel a bit wobbly. Then I go and take a reading and then I've realised I'm like 3.5 reading, [right] and that worried me a little bit,' (Participant K)

'Now I'm just—I'll get on with it like anything else really, but I'll just take in mind that it's something I need to think about when I'm preparing for a session.' (Participant E)

'I've been given numbers to aim for at the start of exercise, so check before you start and if it's about that then go ahead. If it's a bit lower then have a little snack of something. I've got quite a lot of information about sport.' (Participant O)

There was a wide spectrum of confidence levels, from those for whom the anxiety around managing their diabetes during activity prevented most physical activities (eg, participant C) to those who had confidence in their ability to manage their blood glucose and concrete plans to increase exercise levels (eg, participant N). The biggest influences on participants' determination to improve activity levels were motivation and confidence. Participants broadly fell into three groups: those confidently building up their activity levels already or who had concrete plans to do so (CONFIDENT), those keen to increase exercise levels but inhibited by their anxieties (mainly relating to diabetes management) (CONCERNED) and those not particularly interested in currently increasing activity levels (AMBIVALENT). Even highly confident participants had concerns about some aspects of diabetes management.

Several factors emerged that may contribute to an individual's confidence levels. The most important to the majority was information regarding management of diabetes around exercise. In addition, time since diagnosis, experience (both prior experience of exercise and experiences since diagnosis) and confidence in and communication with HCPs were also important. Many participants mentioned information and education about blood glucose management during exercise in this context.

While many participants felt they had received inadequate information about diabetes management around exercise, some felt they had got all the information needed and one felt they had more than enough information.

Information people said they needed ranged from which exercises were suitable for someone with diabetes and which to avoid, to what to expect with blood sugars during exercise, to information on the benefits of exercise to people with T1DM.

'Yeah I wasn't aware, I thought that, as soon as I did exercise it would happen immediately as well, that my sugars would drop and then I'd go funny—so I'd thought I'd be fine the first time I went to the gym … and then a couple of hours later I'd had a hypo, as I didn't realise. Nobody told me that that would happen as well.' (Participant B)

'Erm so yeah, as I say, if I was better informed about what exercise could do to blood sugar levels, then maybe I'd have got back into it quicker.' (Participant H)

'I need more explanation of—into things, what you can do and what you can't do.' (Participant C)

'Educating them that they understand the benefits of exercise; that maybe will encourage them to do it, really.' (Participant M)

Prior experience of exercise and experiences of exercise since T1DM diagnosis could either positively or negatively impact on participants' confidence. For example, participants with previous positive experiences of exercise (eg, D, E and N) were more confident than those who had not (eg, M) and those who had experienced problems with hypoglycaemia or performance since diagnosis (eg, B) were also less confident.

The participants' relationship with their HCPs was important, some getting a lot of support and information (eg, N, O), others having negative experiences such as being advised not to exercise (B, C, D), information about activity and blood glucose management not being forthcoming (B) and getting different messages about diabetes from different HCPs (eg, generalist versus specialist personnel) (K).

Several participants felt that information/knowledge about how to manage diabetes during exercise was out there but just not accessible.

'Information. Because I mean Olympic athletes are doing it, so they must have some kind of regulatory system that they know about that helps you while you're exercising. I mean that would be helpful to disseminate that information' (Participant D)

'I mean like yeah, if, if there was some like, you know, stuff like perfect rule book for if you do X amount of this type of exercise, you know, your blood sugar might be changing by such amount, or something like that.' (Participant H)

### Suggestions to improve activity levels

Participants suggested a number of ways to improve activity levels. A few felt they would not need further encouragement or motivation as they had plans in place. Ideas included additional education, supervised or group activity sessions, a programme of gradually increasing exercise, help with goal setting and a fitness advisor. Although some participants mentioned cost as a potential barrier, nobody felt assistance with this would be particularly helpful.

### Educational material

Nearly all participants felt education about diabetes management was vital in helping improve exercise levels. Some felt they needed more than they had already been given, while others felt they had all they required but this had been important. Participants most confident about increasing activity levels tended to be happier about the information they had received.

'some kind of health organisation to kind of bring forward a website or pamphlet or whatever about people who want to do sports with diabetes type 1 or even diabetes type 2 now and how to deal with certain things and prepare for them.' (Participant E)

Some participants (eg, Participant F) felt overwhelmed by the information they had already been given (although this had not specifically included management of diabetes during exercise), did not currently want further information, but thought it might be useful in the future. Others were happy with the timing of their education or would have preferred more information sooner.

### Supervised or group exercise

Many participants suggested an exercise group, with other patients with T1DM or supervised exercise sessions, with staff with T1DM training. Having a trainer with specific T1DM expertise was important to most, as several participants had experienced ill-informed remarks from members of the public, however, generally it was not felt an HCP was necessary. One person suggested although specific expertise in the trainer was desirable, if there was easy access to advice from the healthcare team, it may not be required. The proposal of group activity sessions was not universally liked and was rejected by some, who preferred exercising under their own steam.

'My dad had a heart attack last year and he got help from the hospital and the hospital gym and he was monitored in a way that he could feel confident with going and doing exercise and helping him—help his heart and diabetics don't get that.' (Participant B)

### Fitness advisor

Regular contact with a fitness advisor, particularly one with T1DM knowledge, was suggested by some as a potential motivator to improve activity levels. Even participants who were happy setting their own programme and targets

felt regular checks would not be unhelpful. Some participants wanted specific advice on a training programme, while others wanted the regular contact and reassurance of someone with greater experience advising them. For some participants, it was very important the advisor could guide them on diabetes management as well as exercise training.

'So you could see a nurse at the hospital or see like a fitness erm—fitness expert at a gym because then you're actually at the place you're going to do it, and you're seeing everybody else doing it, so you might go 'I'll do it'. (Participant D)

### Gradual introduction of exercise

Advice on types of activities and how to build this up was suggested as potentially helpful by some. Others, generally those with previous experience of exercising successfully, felt it was unnecessary. In addition, most welcomed the idea of someone checking on their progress and thought they would find this motivating.

'I probably would want advice of how if they say I want you to increase erm from 30 min walking to an hour walking, or to doing abs in the gym from half an hour to 30 min, yeah,' (Participant K)

### Targets

On a similar note, in general, participants felt target setting would motivate them to increase their exercise particularly if there was a regular check on progress with an advisor.

'I find targets very helpful because I know then—I know what I have to try and get to—I know I have to try and [hmm] reach really. [yeah] It's a bit of competition as well.' (Participant J)

### Monitoring of exercise during a trial

Although most participants were not familiar with the use of an accelerometer to monitor activity levels, no one felt their use during a trial would be onerous. All participants stated they would be happy to keep a diary of their activities and would use a heart rate monitor.

### DISCUSSION

This is the first qualitative interview study to examine attitudes and barriers to exercise in patients newly diagnosed with T1DM. We have identified five themes discussed by patients when they are asked about exercise levels. These are: existing attitudes to exercise; feelings about diagnosis; perceptions about the consequences of exercise; barriers to increasing exercise; confidence in managing blood glucose.

Around half of participants reported a decline in activity levels around the time of diagnosis. This is an important finding, as if it is true of the wider T1DM

population and not addressed, patients may be less willing to be active than the general population. It is reassuring that participants wished to increase their exercise levels as a way to improve their health after a T1DM diagnosis. It is possible that following diagnosis, patients are keen to improve their lifestyle, as is seen in studies of cancer survivors,[14][15] making use of the 'teachable moment'.

In general, exercise was felt to positively impact on health. Some participants were unsure of the benefits or concerned they may harm themselves through exercise. These concerns could be addressed by HCPs during diabetes education.

Many of the barriers identified here have been previously identified in healthy people, as well those with other chronic diseases including longstanding T1DM.[7][16–21] However, our interviewees placed greater emphasis on fear of hypoglycaemia than previous studies of patients with longstanding T1DM.[9] Furthermore, the finding that some patients with diabetes are being advised not to exercise by HCPs has not been previously identified in T1DM qualitative studies and was cited by participants from three different sites.

This study identifies a number of ways in which improvement in exercise levels might be facilitated in patients newly diagnosed with T1DM. In this group particularly, it is critical that confidence in managing diabetes around exercise is addressed. Some interventions identified in this study that may improve confidence in patients newly diagnosed with T1DM and facilitate improved exercise levels were: consistent advice from HCPs; support from diabetes teams for exercise; patient education and time to adjust to diagnosis.

Participants were frustrated by receiving conflicting advice and incorrect information from HCPs. They expected them all to have a basic level of knowledge about diabetes, and this expectation is not being met.

Those who were successfully exercising reported getting strong support from their diabetes team. It is difficult to say whether this was the reason for their success or whether because they were exercising they obtained the information that they required. It was suggested knowledge and support was not forthcoming unless brought up by the patient. Diabetes teams should more positively encourage exercise from diagnosis.

Lack of confidence in managing blood glucose levels around exercise was attributed to a lack of information by most people. Patient resources about blood glucose management around exercise are scarce and although several participants reported searching for these, only one had actually been given any written information. Information on the benefits of exercise in diabetes would have been valued by a majority of study participants. A number of participants talked about the number of appointments they had to attend since diagnosis, the fact they were constantly injecting insulin and checking their blood sugar. Their priority was to 'get their diabetes right' before adding more complexity into the mix. Some patients need more time than others to adjust to their illness.

## Strengths and weaknesses

This study describes the attitudes to exercise of patients recently diagnosed with T1DM, the first qualitative interview study to do so. Due to the fact that we only interviewed them at one time point, we are unable to comment on any causal associations between recent diabetes diagnosis and changes in exercise behaviours. Recruitment was from multiple UK sites, covering both large teaching and district general hospitals, and participants spanned a wide age range. Numbers of participants in qualitative studies vary widely and it is important that saturation of the data is achieved, as it was in this study.

Due to slow recruitment, data were collected in different ways (individual interviews, a focus group, face-to-face and by telephone). Our results should be interpreted with this in mind as participants may be more forthcoming in some of these environments than others. This may have affected individual responses, but could also contribute to a greater breadth of data acquired.[22]

It is likely however that study participants were more interested in exercise than those who declined, and interest in exercise education and management of diabetes around exercise may be lower in the general clinic population. It is possible that patients in other geographical areas and women (who were less well represented in this study) may have different views to those reported here.

## CONCLUSION/RECOMMENDATIONS

Exercise should be encouraged (not discouraged) from diagnosis, as possibly at this time, patients are more amenable to lifestyle change. Advice, particularly on managing insulin doses and carbohydrate intake around exercise, needs to be available both to HCPs and patients with T1DM so that we can help patients to develop confidence managing their diabetes both generally and around exercise. A consensus statement has been published on exercise management in T1DM[23] and based on these guidelines we are developing an education programme to guide insulin and carbohydrate adjustment for safe exercise for HCP and patients with T1DM.[24]

**Acknowledgements** Nikki Jackson (University of Bristol), Dylan Thompson and Keith Stokes (University of Bath), Mary Charlton (Queen Elizabeth Hospital, Birmingham). Roger Holder and Sayeed Haque (University of Birmingham). We are also grateful to Dr George Dowswell, University of Birmingham for the significant contribution he made to this work, and who passed away in July 2016 – we are all the lesser for this loss. We gratefully acknowledge the time and effort of patients who have participated in this trial. We would like to thank staff and colleagues at diabetes centres at the following hospitals for their help with the recruitment of patients and with undertaking this study: Queen Elizabeth Hospital Birmingham, Musgrove Park Hospital Taunton, Bristol Royal Infirmary, Southmead Hospital Bristol, Gloucester, Yeovil, Queen Elizabeth II Hertfordshire, Pinderfields Yorkshire, Churchill Oxford, Alexandra Redditch, George Eliot, Russells Hall, Walsall, New Cross Wolverhampton, Heartlands Birmingham, City Birmingham, Weston General, Royal United Bath, Royal Devon and Exeter.

**Contributors** The study was conceived and designed by PN, RCA, AD and SMG. AK carried out the data collection and AK the analysis with support from PN. GD and SMG. AK drafted the initial manuscript and all authors contributed to critically revising further versions of the manuscript.

**Funding** This work was funded by the National Institute of Health Research grant number PB-PG-0609-19093. SMG is part funded by the National Institute for Health Research (NIHR) Collaboration for Leadership in Applied Health Research and Care West Midlands (CLAHRC WM).

**Disclaimer** The views expressed are those of the authors and not necessarily those of the NIHR, the NHS or the Department of Health.

**Competing interests** None declared.

**Patient consent** Not required.

**Ethics approval** The study received ethical opinion approval from Birmingham, East, North and Solihull Research Ethics committee in February 2010 (reference number 10/H1206/4). The study was sponsored by the University of Birmingham.

**Provenance and peer review** Not commissioned; externally peer reviewed.

**Data sharing statement** The authors confirm that all data underlying the findings are fully available without restriction. All relevant data are within the paper.

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
