## [Reviewer comments · BMJ Open]

ARTICLE DETAILS

TITLE (PROVISIONAL)	Attitudes and barriers to exercise in adults with a recent diagnosis of type 1 diabetes: a qualitative study of participants in the Exercise for Type 1 Diabetes study.
AUTHORS	Kennedy, Amy; Narendran, P; Andrews, Robert; Daley, Amanda; Greenfield, Sheila

VERSION 1 – REVIEW

REVIEWER	Dr Christine P Burren University Hospitals Bristol NHS Foundation Trust UK
REVIEW RETURNED	01-Jun-2017

GENERAL COMMENTS	This study is an informative study on the thoughts and understanding of adults newly diagnosed with diabetes. This qualitative study provides valuable insight to their views on the topic of exercise. 'Exercise', however is a broad group of very different activities. The level of description was too superficial on this front. More detail should be provided on types of exercise already undertaken by the participants. The researchers state that some were considering increasing exercise now that they had diabetes, again what type? The relevance is that is unclear if the exercise types that participants refer to would fit into the Moderate, Moderately Vigorous or Vigorous categories of accelerometer data activity levels, which have an impact on the subsequent study. eg for the participants, does exercise mean walking, running, cycling, rock climbing or weights at the gym? These different exercise types are also likely to have different impact on hypoglycaemia risk and prevention strategies. This information should exist for inclusion, as the accompanying EXTOD protocol outlines in the Topic Guide for these Phase 1 interviews ie activity and exercise behaviour – 'What exercise do you do?' 'What does the mean exercise mean to you?' The second area to be addressed is that they abstract outlines one aim of exploring 'preferences for monitoring of activity in a trial' - this is not addressed within the manuscript. It did not emerge within any of the five themes identified in the study (page 5). Whilst many of the participants' statements provide insight and will be of great value to the researchers in their next study phase, this aim was not directly fulfilled within this current study. Further clarification is needed - either it should be removed as an aim, or they should then state that that aim was not fulfilled, or additional detail addressing this aim should be included.
---

	Although the study includes 15 people and thus statistics on responses on not appropriate, some form of numerical description would be appropriate to accompany the 'many' and some' throughout, as these are quite subjective phrases. Accompanying numbers would give better description for the reader. P12: 'many' and 'some' would be valuable to put a number to these instead. Minor points. P5 line 20 mentions 19 sites yet abstract mentions 15. Clarification of wording needed. P5 line 20 there seems to be a stray comma.
--	---

REVIEWER	Elizabeth Beverly Ohio University Heritage College of Osteopathic Medicine, USA
REVIEW RETURNED	07-Jul-2017

GENERAL COMMENTS	Strengths: I commend the authors on a well-written manuscript addressing newly diagnosed type 1 diabetes patients' attitudes and barriers to exercise. The authors' conducted a thorough analysis of their qualitative data. More research is needed to confirm these findings in order to develop exercise interventions and educational materials for healthcare professionals and patients newly diagnosed with type 1 diabetes. Revisions: Methods: The distribution of the study sample is skewed towards men with type 1 diabetes from Bristol. Did the authors consider recruiting more women as well as patients from the different centres to ensure that saturation of the data was achieved? It is possible that women and patients from different regions of the country have different views toward exercise. Methods: The methods section should include a subsection on "Rigor" or "Efforts to Ensure Quality." The emergent nature of qualitative research necessitates special attention to address issues of rigor in research methods and investigations. To strengthen the findings, the authors should address these issues. Rigor can be supported by tangible evidence including audit trails, member checks, memos, etc. For example, the authors may want to consider including the interview questions to enhance transferability (i.e., external validity). The questions may help the reader understand the mapping process and themes. Results: In the Themes section, the authors include more descriptive theme titles in parentheses. I recommend the authors use the more descriptive titles for the subheadings of each theme. Results: The authors included few, brief quotations in the text for the themes. Quotations support the researchers' claims and illustrate the meaning of themes. The lack of quotations as well as the brevity of quotations may be indicative of a lack of adequacy and/or appropriateness of the data. In other words, the authors may not have adequate data (or volume) to support the emergent themes or quality of data to provide the descriptive and interpretive depth required to clearly delineate participants' views.
---

	The authors should consider incorporating additional, lengthier quotations in the text for each theme. Discussion: The authors need to address additional limitations to the study, including the small sample size (even for a qualitative study), the small number of female participants, the wide age range among participants, and the homogeneity of the study sample with regards to race/ethnicity. Also, the cross-sectional study design limited the ability to detect any causal associations between recent diabetes diagnosis and changes in exercise behaviors. Finally, the authors need to address potential limitations from the multiple modes of data collection (i.e., conducting one focus group, in-person interviews, and telephone interviews) and rigor. Conclusions: The authors need to expand upon their conclusions/recommendations. For example, what types of advice do HCPs and T1DM patients need? Concrete examples would be very helpful to clinicians, educators, and researchers who are reading this manuscript.
--	---

VERSION 1 – AUTHOR RESPONSE

Reviewer 1

This study is an informative study on the thoughts and understanding of adults newly diagnosed with diabetes. This qualitative study provides valuable insight to their views on the topic of exercise.

Thank you very much for these comments.

Comment: 'Exercise', however is a broad group of very different activities. The level of description was too superficial on this front. More detail should be provided on types of exercise already undertaken by the participants. The researchers state that some were considering increasing exercise now that they had diabetes, again what type? The relevance is that is unclear if the exercise types that participants refer to would fit into the Moderate, Moderately Vigorous or Vigorous categories of accelerometer data activity levels, which have an impact on the subsequent study. eg for the participants, does exercise mean walking, running, cycling, rock climbing or weights at the gym? These different exercise types are also likely to have different impact on hypoglycaemia risk and prevention strategies. This information should exist for inclusion, as the accompanying EXTOD protocol outlines in the Topic Guide for these Phase 1 interviews ie activity and exercise behaviour – ‘What exercise do you do?’ ‘What does the mean exercise mean to you?’

Response; We have added the following sentences to the article and also include a table of their activity levels

“Activities that participants classed as exercise varied from walking during their working day to swimming or going to the gym. Table 2 shows the exercise that each participant was taking part in. 10 participants were doing moderate activity, 1 moderately vigorous activity and 3 vigorous activity.”

Comment: The second area to be addressed is that they abstract outlines one aim of exploring 'preferences for monitoring of activity in a trial' - this is not addressed within the manuscript. It did not emerge within any of the five themes identified in the study (page 5). Whilst many of the participants' statements provide insight and will be of great value to the researchers in their next study phase, this aim was not directly fulfilled within this current study. Further clarification is needed - either it should be removed as an aim, or they should then state that that aim was not fulfilled, or additional detail addressing this aim should be included

Response: Thank you for highlighting this. We have added a section about this (page 15).

“Monitoring of exercise during a trial

Although most participants were not familiar with the use of an accelerometer to monitor activity levels, no-one felt their use during a trial would be onerous. All participants stated they would be happy to keep a diary of their activities and would use a heart rate monitor.”

Although the study includes 15 people and thus statistics on responses on not appropriate, some form of numerical description would be appropriate to accompany the 'many' and 'some' throughout, as these are quite subjective phrases. Accompanying numbers would give better description for the reader. P12: 'many' and 'some' would be valuable to put a number to these instead.

Response:

There has been huge debate in the field about whether numbers should be used in reporting qualitative research. We have elected not to use them here for the following reasons

1. Numbers can lead to the inference by the reader of greater generality for the conclusions than is justified.
2. The use of numbers can lead to the reader a sliding into variance way of thinking something we try to avoid in qualitative research.
3. In reporting the numbers there is a danger of reducing evidence to the amount of evidence.

We have added a sentence and a supporting reference stating this.

“Specific numbers of participants are not routinely given throughout as these are not generally used when reporting qualitative research, the aim of sampling being to represent the spread of views rather than proportions which can be generalised to a larger group (ref 13 - Maxwell JA. Using Numbers in Qualitative Research. Qualitative Inquiry. 2010;16(6):475-482). “

Comment: P5 line 20 mentions 19 sites yet abstract mentions 15. Clarification of wording needed.

We have changed the Abstract to say 19

Comment: P5 line 20 there seems to be a stray comma.

Thank you we have corrected this

Reviewer 2

Comment: Strengths

I commend the authors on a well-written manuscript addressing newly diagnosed type 1 diabetes patients' attitudes and barriers to exercise. The authors' conducted a thorough analysis of their qualitative data. More research is needed to confirm these findings in order to develop exercise interventions and educational materials for healthcare professionals and patients newly diagnosed with type 1 diabetes.

Response: Thank you for these comments

Comment:

Methods

The distribution of the study sample is skewed towards men with type 1 diabetes from Bristol. Did the authors consider recruiting more women as well as patients from the different centres to ensure that saturation of the data was achieved? It is possible that women and patients from different regions of the country may have different views toward exercise.

Response: We did not do purposeful sampling, instead we tried to recruit from the 19 centres that we had open for the EXTOD study. This study sought views from patients newly diagnosed with Type 1 diabetes. Diabetes is a rare disease with an incidence of 1 in 5000 and occurs almost twice as common in men (1.8:1 – male to female). This means that centres that covered a greater population would recruit more patients and there would likely be more men recruited. Bristol and Birmingham are the centres that covered the greatest population hence why there are more interviewees from there. We did manage to get participants from three other centres, Wakefield (North England), Gloucester (Mid England) and Taunton (South England) so feel that we had reasonable geographic cover. Men are more likely to come into an exercise study so this did mean that our ratio of men to women was higher than seen by diagnosis (2.7:1 compared to diagnosis rates of 1.8:1). We did not look for gender specific saturation, but rather saturation for the whole group and stopped when we reached this. The women did not bring up any additional themes that were not brought up by the men (IS THIS TRUE) but on reflection agree that we should have looked for female and male saturation. We have added this as shortcoming of this paper.

“It is possible that patients in other geographical areas and women (who were less well represented in this study) may have different views to those reported here.”

Methods

Comment: The methods section should include a subsection on “Rigor” or “Efforts to Ensure Quality.” The emergent nature of qualitative research necessitates special attention to address issues of rigor in research methods and investigations. To strengthen the findings, the authors should address these issues. Rigor can be supported by tangible evidence including audit trails, member checks, memos, etc. For example, the authors may want to consider including the interview questions to enhance transferability (i.e., external validity). The questions may help the reader understand the mapping process and themes.

The COREQ: 32-Item Checklist (designed with quality and transparency in mind) reports in detail on the various processes used in the study. As required by the journal reporting guidelines this was submitted with the article and available to readers it is stated that this is published with the final article

Results

Comment: In the Themes section, the authors include more descriptive theme titles in parentheses. I recommend the authors use the more descriptive titles for the subheadings of each theme.

Response: We have used the more descriptive titles.

Results

Comment: The authors included few, brief quotations in the text for the themes. Quotations support the researchers' claims and illustrate the meaning of themes. The lack of quotations as well as the brevity of quotations may be indicative of a lack of adequacy and/or appropriateness of the data. In other words, the authors may not have adequate data (or volume) to support the emergent themes or quality of data to provide the descriptive and interpretive depth required to clearly delineate participants' views. The authors should consider incorporating additional, lengthier quotations in the text for each theme.

Response: We have added additional quotes in each theme.

Discussion

Comment: The authors need to address additional limitations to the study, including the small sample size (even for a qualitative study), the small number of female participants, the wide age range among participants, and the homogeneity of the study sample with regards to race/ethnicity.

Response: We have added a sentence about this. "It is possible that patients in other geographical areas and women (who were less well represented in this study) may have different views to those reported here."

Also, the cross-sectional study design limited the ability to detect any causal associations between recent diabetes diagnosis and changes in exercise behaviors.

There is currently no way of accurately detecting who will get Type 1 diabetes in the next year so it not possible to conduct interviews before and after diagnosis. Interviewing at diagnosis and at a later date may give some insight into the causative associations between recent diabetes diagnosis and changes in exercise behaviors. This study though did not set out to answer this questions. We hope in an up and coming study to be able to do that. We have added a line to highlight that we are unable to comment on causality.

"Due to the fact that we only interviewed them at one time point we are unable to comment on any causal associations between recent diabetes diagnosis and changes in exercise behaviors"

Comment: Finally, the authors need to address potential limitations from the multiple modes of data collection (i.e., conducting one focus group, in-person interviews, and telephone interviews) and rigor.

Response: We have added a paragraph addressing this together with a supporting reference .
(P. Gill, K. Stewart, E. Treasure, B. Chadwick Methods of data collection in qualitative research: interviews and focus groups. British Dental Journal 204, 291 - 295 (2008)
Published online: 22 March 2008 | doi:10.1038/bdj.2008.192)

"Due to slow recruitment, data were collected in different ways (individual interviews, a focus group, face-to-face and by telephone). Our results should be interpreted with this in mind as participants may be more forthcoming in some of these environments than others. This may have affected individual responses, but could also contribute to a greater breadth of data acquired."

Conclusions

Comment: The authors need to expand upon their conclusions/recommendations. For example, what types of advice do HCPs and T1DM patients need? Concrete examples would be very helpful to clinicians, educators, and researchers who are reading this manuscript.

Response: The purpose of this study is not to identify the advice that should be given. We have though highlighted the reader to a recent consensus on what advice should be given and also to an education programme that we are developing. The conclusion has changed from

“Exercise should be encouraged (not discouraged) from diagnosis, as possibly at this time, patients are more amenable to lifestyle change. Advice needs to be made available both to HCPs and T1DM patients so that we can help patients to develop confidence managing their diabetes both generally and around exercise.”

To

“Exercise should be encouraged (not discouraged) from diagnosis, as possibly at this time, patients are more amenable to lifestyle change. Advice, particularly on managing insulin doses and carbohydrate intake around exercise, needs to be available both to HCPs and T1DM patients so that we can help patients to develop confidence managing their diabetes both generally and around exercise. A consensus statement has been published on exercise management in Type 1 diabetes²³ and based on these guidelines we are developing an education programme to guide insulin and carbohydrate adjustment for safe exercise for HCP and T1DM patients²⁴.”

VERSION 2 – REVIEW

REVIEWER	CP Burren University Hospitals Bristol NHS Foundation Trust Bristol UK No Competing Interest
REVIEW RETURNED	27-Sep-2017
GENERAL COMMENTS	Amendments appropriate, giving a more balanced picture.